# Resource landscape shapes the composition and stability of the human vaginal microbiota

Tsukushi Kamiya[1]*, Mircea T. Sofonea[2,3], Michael France[4], Nicolas Tessandier[5], Ignacio G Bravo[5], Carmen Lia Murall[5,6], Jacques Ravel[4], Samuel Alizon [1]*

1 Center for Interdisciplinary Research in Biology, Collège de France, CNRS, INSERM, Université PSL, Paris, France, 2 PCCEI, Univ Montpellier, INSERM, Univ Antilles, Montpellier, France, 3 Department of Anesthesiology, Critical Care, Intensive Care, Pain and Emergency Medicine, Centre Hospitalier Universitaire de Nîmes, Nîmes, France, 4 Institute for Genome Sciences, Department of Microbiology and Immunology, University of Maryland School of Medicine, Baltimore, Maryland, United States of America, 5 MIVEGEC, CNRS, IRD, Université de Montpellier, Montpellier, France, 6 National Microbiology Laboratory, Public Health Agency of Canada, Montréal, Canada

* tsukushi.kamiya@college-de-france.fr (TK); samuel.alizon@college-de-france.fr (SA)

## Abstract

The vaginal microbiota is associated with the health of women and newborns alike. Despite its comparatively simple composition relative to other human microbiota systems, the mechanisms underpinning the dynamics and stability of vaginal microbial communities remain elusive. A crucial, yet so far underexplored, aspect of vaginal microbiota ecology is the role played by nutritional resources. Glycogen and its derivatives, produced by vaginal epithelia, are accessible to all bacterial constituents of the microbiota. Concurrently, free sialic acid and fucose offer supplementary nutritional resources to bacterial strains capable of cleaving them from glycans, which are structurally integral to mucus. Notably, bacteria adept at sialic acid exploitation are often correlated with adverse clinical outcomes and are frequently implicated in bacterial vaginosis (BV). In this study, we introduce a novel mathematical model tailored to human vaginal microbiota dynamics to explore the interactions between bacteria and their respective nutritional landscape. Our resource-based model examines the impact of the relative availability of glycogen derivatives (accessible to all bacterial species) and sialic acid (exclusive to some BV-associated bacteria) on the composition of the vaginal microbiota. Our findings elucidate that the success of BV-associated bacteria is intricately linked to their exclusive access to specific nutritional resources. This private access fortifies communities dominated by BV-associated bacteria, rendering them resilient to compositional transitions. We empirically corroborate our model prediction with longitudinal clinical data on microbiota composition and previously unpublished metabolomic profiles obtained from a North American cohort. The insights gleaned from this study shed light on potential pathways for BV prevention and treatment.

**Data availability statement:** The supporting information PDF file (S1 Text) contains the Extended Methods and Supplementary Results. All the data and scripts used to generate the figures can be accessed at https://doi.org/10.57745/3GJF2Z.

**Funding:** Research reported in this publication was supported by the Fondation pour la Recherche Medicale (award SPF202005011951 to TK), the Expos'UM institute (to NT) and also in part by the National Institute for Allergy and Infectious Diseases (awards R01NR015495 to JR) and the National Institute of Nursing Research of the National Institutes of Health under (award UH2AI083264 to JR), the Gates Foundation (award OPP1189205 to JR), and the CUPS2 project from the Agence Nationale de la Recherche (award ANR-22-CE34-0024 to SA). The funders had no role in study design, data collection and analysis, decision to publish, or preparation of the manuscript.

**Competing interests:** We have read the journal's policy and the authors of this manuscript have the following competing interests: JR is the cofounder of LUCA Biologics, a biotechnology company focusing on translating microbiome research into live biotherapeutics drugs for women's health, and a member of Ancilia Bio Scientific Advisory Board. The other authors have declared that no competing interests exist.

## Significance statement

The vaginal microbiota has a notable impact on women's health at various stages of life, namely puberty, infection protection, sexual health, fertility, pregnancy, and menopausal changes. At present, most non-anti-microbial products developed to mitigate adverse vaginal symptoms emphasise competitive interactions through acids (boric or lactic acid) or probiotics as a means to "rebalance" microbiota communities. Despite recent advances in profiling the composition of vaginal microbiota communities, there remains a major gap in our mechanistic understanding of how to maintain or reinstate a resilient *Lactobacillus*-dominated microbiota that improves vaginal health and outcomes. This study explores the role of nutritional resources in the vaginal microbiota by introducing a mathematical model that analyses how access to specific nutrients, like glycogen derivatives and sialic acid affects the balance of bacterial vaginosis (BV) and non-BV-associated bacteria. Our findings, supported by original cohort-derived microbiological and metabolomics data, demonstrate that exclusive access to these nutrients is linked to the dominance and resilience of BV-associated bacteria, providing new insights for BV prevention and treatment.

## Background

The integration of community ecology theory in human microbiota studies remains a central goal, as ecological processes that shape microbial community composition hold the key to understanding the role of microbiota in health and disease [1]. Gut microbiota garners most of the attention among human microbiota systems. Nevertheless, the application of traditional ecological theories to the intestinal milieu continues to be generic in scope. This is in part due to the vast diversity of these communities, which display between-individual variability, temporal composition fluctuations, as well as within-host spatial diversity in different regions of the digestive tract.

While detailed mechanistic understanding of the human gut microbiota ecology remains elusive, ecological theory has been increasingly applied to the dynamics in chemostat bioreactors, in which defined bacterial communities are cultured under controlled settings [2–5]. Under such *in vitro*, simplified conditions, mathematical models can describe interactions between bacteria and their environment to uncover factors shaping the composition of these synthetic microbial communities. Nonetheless, the extent to which these findings from *in vitro* settings scale up to *in vivo* microbial communities and, more generally, how ecological theory can improve our understanding of human microbiota dynamics remains an open question. In this work, we focus on human vaginal microbiota communities, whose relative simplicity makes them well-suited for applying ecological theory to a real-world setting and elucidate mechanisms underlying health and dysbiosis, understood here as imbalances or deviations from eubiotic and optimal states [6].

Defining dysbiosis is essential for establishing a shared understanding of what microbial community compositions are considered beneficial or harmful. Yet this remains particularly challenging in the gut microbiota since the classification—and even the existence itself—of distinct microbiota community types in the gut (known as 'enterotypes') continues to be contentious [7]. By contrast, the strong structuring of

the vaginal microbiota into community state types, or CSTs, is widely recognised and has been well-documented in association with health and dysbiosis [8,9]. Briefly, the human vaginal microenvironment is most often dominated by one of four *Lactobacillus* species [8,10]: CST I, dominated by *L. crispatus*; CST II, dominated by *L. gasseri*; CST III, dominated by *L. iners*; and CST V, dominated by *L. jensenii*. CSTs I, II, and V are often regarded as 'optimal', offering protection against disease and adverse health effects, whereas CST III plays a more ambiguous or mixed role [11]. In contrast, in CST IV, lactobacilli give way to a diverse array of strict and facultative anaerobes from the genera *Gardnerella*, *Fannyhessea*, or *Prevotella*. This microbial composition has been consistently linked to adverse gynaecological, obstetric, and mental health outcomes, an increased risk of sexually transmitted infections [12] as well as *in vitro* fertilisation failure [13]. Consequently, CST IV is often regarded as a 'non-optimal' or dysbiotic vaginal microbiota state. This state is typically present in bacterial vaginosis (BV), a condition experienced by approximately 23 to 29 % of women of reproductive age, depending on the region [14], and characterised by symptoms such as vaginal odour and discharge, and associated with severe discomfort [15]. The reverse is not necessarily true, as many women with CST IV do not exhibit any symptoms [15,16]. Despite the relative tractability and well-established health implications of the vaginal microbiota, the application of ecological theory to understanding this dynamic system remains limited.

Here, we adopt a resource ecology perspective, focusing on the role of shared and private nutritional resources in shaping the microbiota composition. Briefly, the nutritional resource landscape of the vaginal environment is described as follows. The human vagina is lined with stratified layers of epithelial cells, which are renewed and shed from the basal and outer layers, respectively. Epithelial cells store carbohydrates in the form of glycogen, which may be released to the extracellular milieu by shedding outer epithelial cells and through the destruction of immature cells (i.e., cytolysis) by *G. vaginalis* and other CST IV bacterial species [17]. The degree to which each mechanism contributes to the availability of extracellular glycogen is unclear. However, the strong negative association observed between pH and cell-free glycogen suggests that glycogen release is driven predominantly by a CST IV independent mechanism [18]: as *G. vaginalis* dominance is associated with elevated pH, one would expect a positive association if cytolysis were a dominant driver. Once released by shedding outer epithelial cells, glycogen and its breakdown derivatives (e.g., glucose, maltose, maltotriose, maltopentaose, and maltodextrins) constitute the primary fuel for vaginal bacterial growth, and they are utilised by both CST I and IV bacteria [19,20]. In addition to glycogen, CST IV bacteria are capable of metabolising additional resources. Specifically, *G. vaginalis* can cleave off sialic acid and fucose molecules from large sugar molecules that comprise the vaginal mucus layer (i.e., mucus sialoglycan) [21,22]. As the CST I does not appear to metabolise the end products of mucus degradation [22], we can consider sialic acid and fucose private to CST IV.

Our mathematical model demonstrates that the relative frequency of BV-associated bacteria hinges upon the relative supply of resources private to them (sialic acid and fucose from the mucus), and that the access to private resources makes communities dominated by BV-associated bacteria resistant to transition. By leveraging vaginal microbiota compositional data and original metabolomic data longitudinally collected from a cohort in North America, we corroborate the role of nutritional resources in shaping vaginal microbiota communities. These findings have implications for BV prevention and provide indications for targeted interventions to improve women's sexual and reproductive health.

## Modelling vaginal microbiota ecology

Mathematical models are instrumental in community ecology to study the consequences of non-linear interplay between organisms and their environment. A large body of theoretical exploration of microbiota systems has relied on classic competitive Lotka-Volterra models, often focusing on species interactions, while overlooking resource availability and consumption as a key driver of bacterial growth and population size [23–25]. In contrast, the work described here builds upon classic models of consumer-resource theory [26,27], which assumes that resources are the primary drivers of community composition and assemblage.

We adopt a parsimonious approach in modelling the composition of the vaginal microbiota and focus on two communities associated with distinct health outcomes (Fig 1): on the one hand, CST I vaginal microbiota, dominated by *L. crispatus*, that maintains an acidic vaginal micro-environment in which BV-associated bacteria tend to perform poorly; on the other hand, CST IV vaginal microbiota, consisting of an assemblage of strict and facultative anaerobic bacteria, typically including species within *Gardnerella*, *Prevotella*, and *Fannyhessea* genera. Both CST II and CST V (dominated by *L. gasseri* and *L. jensenii*, respectively) are less prevalent and generally considered functionally analogous to CST I [17]. As aforementioned, the association of CST III (dominated by *L. iners*) with a clinical presentation remains an open question and does not feature in our model as a distinct entity. Herein, we use the CST identity to refer to its typical microbial profile and implication on health outcomes: i.e., CST I is characterised by a high prevalence of lactobacilli and is compatible with optimal vaginal physiology, while CST IV is characterised by the presence of strict and facultative anaerobes and is most often associated with adverse clinical presentations [17].

In our work, we track the availability of two types of nutritional resources (Fig 1): first, glycogen and its derivatives, which constitute a universal energy source for all vaginal microbiota bacterial components [20], and, second, sialic acid and fucose molecules, which are available as sialogycans attached to mucins that form mucus. Some common anaerobes can access these resources through the production of sialidases and fucosidases, respectively, while *L. crispatus* is typically unable to leverage these resources for growth [22] as the majority of *L. crispatus* strains do not metabolise sialic acid. Furthermore, the strata formed by the sialic acid-cleaving bacteria, with potential biofilms [28], likely allow them to secure private access to this resource, as well as to the associated glycans.

Finally, we modelled the inhibitory interactions mediated by lactic acid produced by lactobacilli (Fig 1). Acidity contributes to the ecology in the vaginal environment by differentially affecting bacterial activity. Bacteria commonly found in

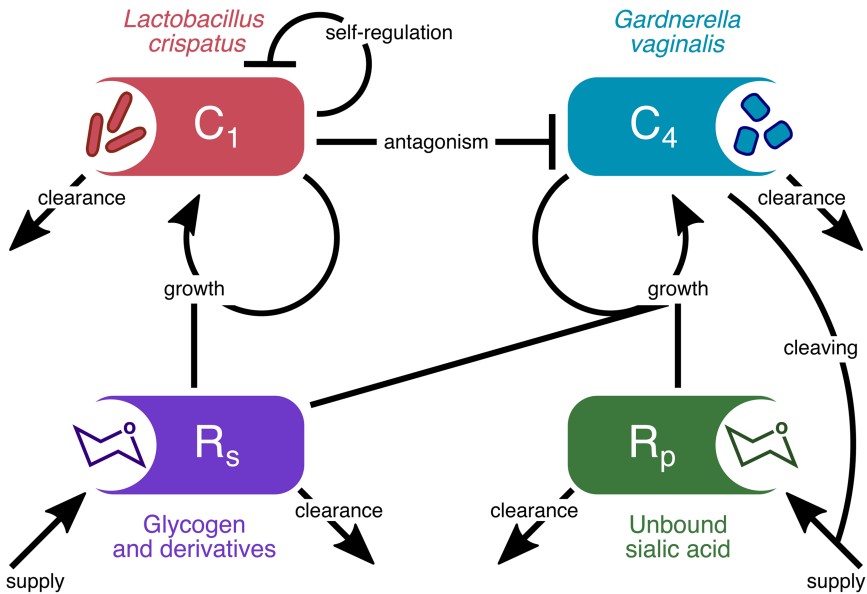

**Fig 1**. **Diagram of the consumer-resource vaginal microbiota model.** We track the dynamics of two vaginal microbiota types (CST I and CST IV associated bacteria; $C_1$ and $C_4$) and the shared ($R_s$) and private resource ($R_p$) utilised by the two communities. The arrows among compartments represent density-dependent interactions including resource-mediated bacterial growth; lactic-acid-mediated density regulation (self-regulation and antagonism); resource supply; sialic acid cleaving; and background clearance of microbes and resources. The representative species and, or molecules for each model component are listed. The corresponding mathematical model based on ordinary differential equations is detailed in the Methods section.

CST IV vaginal microbiota often require a vaginal pH >4.5 to grow and are growth restricted at a lower pH [8,29] while lactobacilli have a higher tolerance to acidic conditions but show growth inhibition at lower pH (<4.0; i.e., self-inhibition) [29, 30]. We note that CST I antagonism towards CST IV in our model represents the net effect of all antagonism mechanisms, including lactic acid and other molecules produced by lactobacilli, such as bacteriocins, as well as the counter-adaptation processes that CST IV bacteria use (e.g., the production of biogenic amines) [17,31].

Changes in the composition and activity of vaginal microbiota can occur in rapid succession in the vaginal environment, including daily fluctuations [32]. Therefore, the model assumes that the resources and bacterial activity reach equilibrium rapidly, and our analysis focuses on steady-state conditions. Our model is based on a system of ordinary differential equations (ODEs): the mathematical formulation and analyses are described in the Methods section.

## Insights from mathematical modelling

We find that the resource landscape of the vaginal microenvironment can explain the composition and structure of the vaginal microbiota. Broadly, our numerical analysis demonstrates that an initially CST I-dominated vaginal microbiota undergoes a transition to CST IV as the proportion of resource supply that is private to CST IV bacteria, $\chi$, increases (Fig 2). When only a small fraction of resources is accessible privately to CST IV, CST I dominates the system at equilibrium since compounds such as lactic acid directly inhibit bacteria associated with CST IV. When more private resources are available to CST IV, it gains dominance, as the benefits from the additional resources eventually outweigh the disadvantages caused by direct suppression from lactobacilli. We also observe an intermediate window of resource condition that allows for bistability (where either state, i.e., of single species-dominance, is stable, depending on the initial state of the system) and the stable coexistence of multiple species (Fig 2; grey shading).

In addition to reducing the supply of private resources to CST IV, our model always requires the (re-)introduction, or external seeding, of CST I-associated bacteria to observe a transition from CST IV to CST I. This was approximated in the model with a non-zero initial density of CST I ($q - 1 = 10^{-10}$ to 0.1; Fig 2). Importation of external microbes to the vaginal milieu may take place through sexual intercourse, contamination from other body parts, and lactobacilli-based probiotics [33–35]. In our model, bistability implies that variable amounts of seeding are necessary to bring the system into either basin of attraction. In particular, if the external seeding of CST I bacteria is insufficient, the CST IV dominance is irreversible (Fig 2a). This finding is consistent with our assumption of a closed system, wherein any increase in the bacterial populations is assumed to be generated solely by bacterial growth, precluding CST I from increasing in frequency if they are initially absent. However, note that a small initial seeding of CST I (e.g., at least 0.1 % of the total initial frequency is made up of CST I despite CST IV dominance) is enough to cross over an unstable equilibrium (i.e., separatrix) and escape the basin of attraction associated with CST IV dominance as long as the resource landscape allows for CST I existence (i.e., sufficiently low $\chi$; Fig 2 b–d).

A larger external seeding of CST I-associated bacteria allows for a stable CST I community to be achieved across a broader range of resource conditions (Fig 2), supporting that external seeding of CST I (i.e., lactobacilli-based probiotics) may represent a promising strategy to restore the lactobacilli-rich community. Conversely, the more the resource landscape is biased toward the needs of CST I (smaller $\chi$; Fig 2), the smaller the perturbation required to establish stability involving CST I bacteria, i.e., away from a stable state dominated by CST IV bacteria ($q = 1$).

We observe that transitions to CST IV can be hysteretic, meaning that recovery from CST IV requires a larger reduction in the CST IV private resources (e.g., sialic acid and fucose) than the initial increase that triggered a transition from CST I to IV. While it is challenging to demonstrate hysteresis in natural settings, hysteretic shifts along a resource gradient have been recently demonstrated in chemostats of intestinal microbes [5].

 

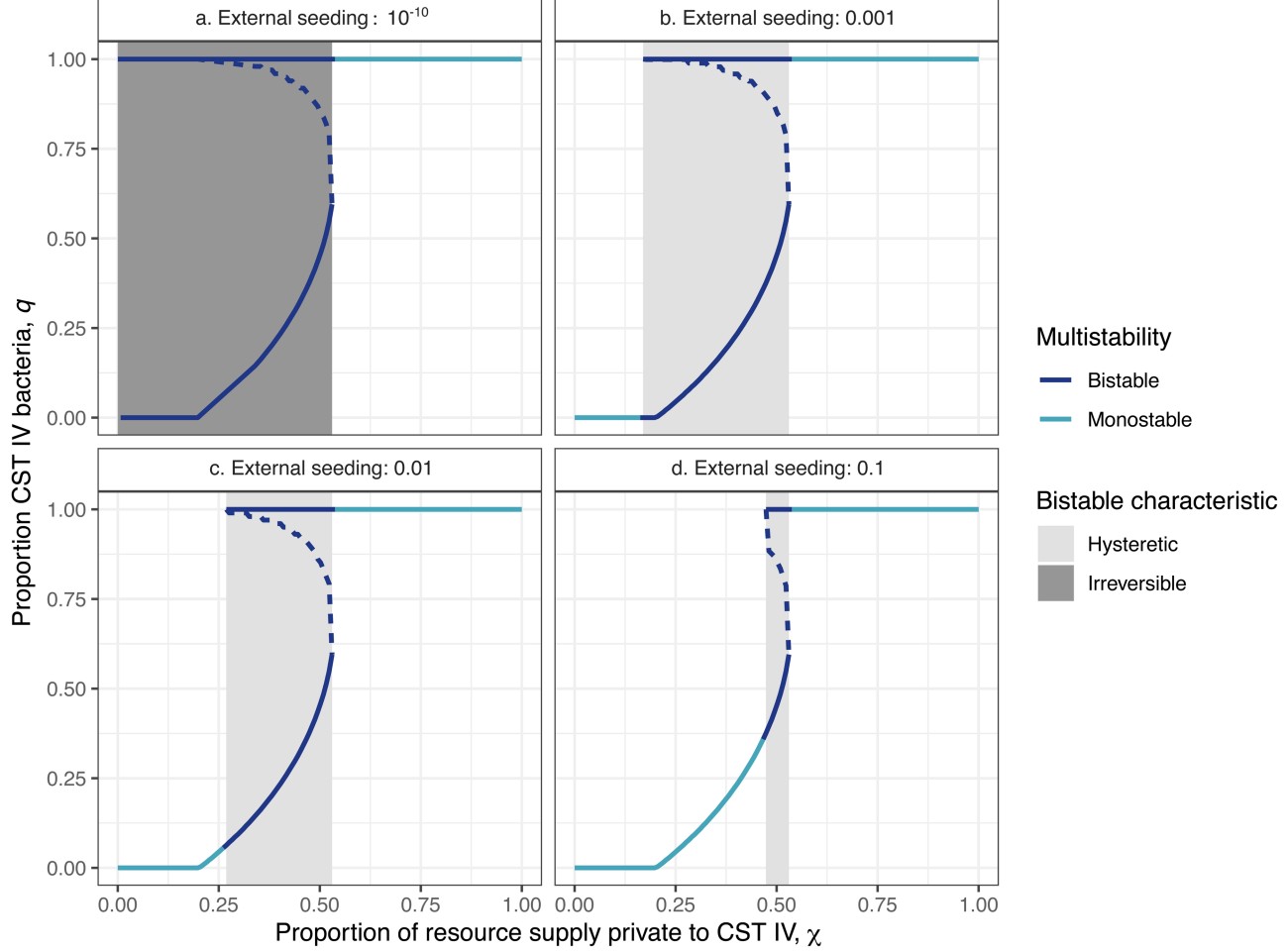

**Fig 2**. **Vaginal microbiota communities undergo a hysteretic or even irreversible transition to CST IV dominance as the proportion of resource supply private to CST IV increases.** Solid lines indicate stable steady states with light blue and dark blue indicating monostable and bistable states, respectively. The dashed lines indicate the separatrix, which separates two basins of attraction, each with a stable state. The derivation of $\chi$ is found in the Extended Methods in S1 Text. All the data and scripts used to generate the figure can be accessed at https://doi.org/10.57745/3GJF2Z.

## Corroborating theory with clinical data

We leverage human vaginal microbiota and metabolomics data to corroborate our theoretical prediction that the relative supply of resources private to CST IV is a driver of the vaginal microbiota composition. A key challenge in interrogating mechanistic model predictions with empirical data is that ecological processes of interest are often difficult or impossible to measure directly. For instance, it is not possible to directly observe the quantity of incoming resources (i.e., rates of glycogen and sialic acid supply). Fortunately, however, the measured quantity of resource-associated metabolites, such as sialic acid and glycogen derivatives (e.g., glucose, maltose, and maltotriose), can yield valuable insights when interpreted through the lens of a mechanistic ecological model. On their own, these metabolites may be misleading as proxies for resource supply because they are influenced by distinct ecological processes, including the resource supply rate, resource processing (e.g., sialic acid and fucose cleavage by CST IV bacteria), and consumption. For instance, a high concentration of unbound sialic acid might reflect either increased input of the bound form or elevated density of CST IV bacteria that release it through enzymatic cleavage. Our model addresses this issue by explicitly delineating the

processes of resource consumption and sialic acid cleavage from resource supply, allowing us to derive an expression for resource supply that is independent of other processes that influence the concentration of unbound sialic acid (eq. 8 in S1 Text): in brief, the key derived quantities, independent of bacterial densities, are the proportion of resource supply private to CST IV ($\chi$) and the overall resource productivity ($\overline{\tau}$). By incorporating these expressions with data on resource-associated metabolites and microbiota composition from a cohort study [32], as well as the relative growth rate of *Gardnerella vaginalis* to *Lactobacillus crispatus* (first figure in S1 Text), we estimate the extent of shared (i.e., glucose, maltose, and maltotriose) and private (i.e., sialic acid) resource supply for each clinical sample.

Analysis of data from a North American cohort of 82 women corroborates that the composition of human vaginal microbiota is associated with variations in resource supply: an increased relative supply of private resources ($\chi$) is associated with increased colonisation by CST IV bacteria (Fig 3a). In particular, at the lowest 10 % quantile of the relative private resource supply, *L. crispatus*-dominated CST I was the most common type (25 out of 46 samples), while the occurrence of CST IV was very rare (1 out of 46 samples). Beyond the 30 % quantile, the CST IV frequency consistently exceeded 50 % (Fig 3a). Our findings were qualitatively identical across a wide range of model assumptions, with the exception of a scenario in which private resources are virtually inaccessible due to sialidase inactivity (Sect 4 in S1 Text).

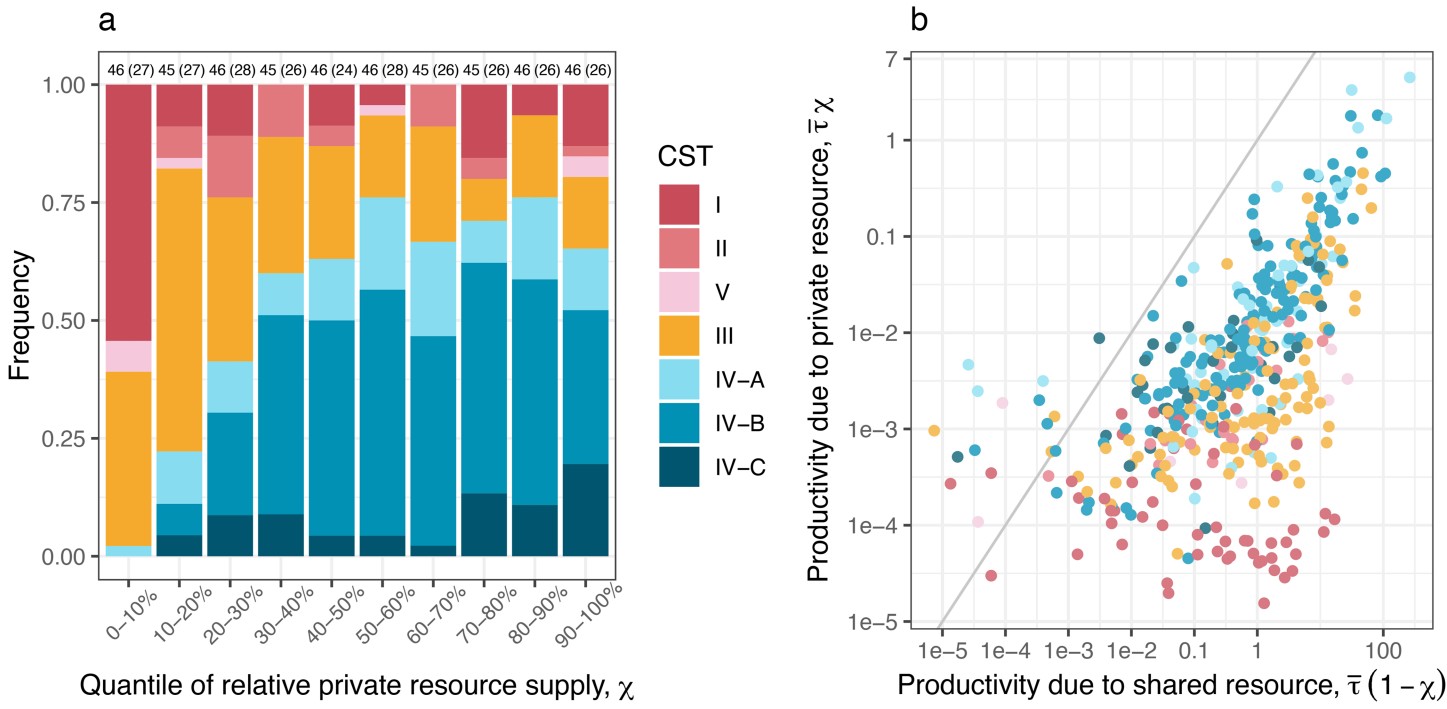

**Fig 3**. **Microbiota and metabolomics data support our model prediction that resource availability shapes human vaginal microbiota structure.** a) Higher supplies of private resources more frequently support CST IV-dominated communities. Shown are the frequencies of CSTs as a function of quantiles of empirically estimated $\chi$, the proportion of supplied resource private to CST IV (i.e., informed by metabolomics and microbiological data). The numbers on top of each bar indicate the number of samples (and women in parentheses), respectively. The frequency of notable vaginal microbiota species as a function of $\chi$ quantiles is shown in the third figure in S1 Text. b) Extremes in private resource availability lead to the exclusion of distinct community types—CSI I excluded under high supply and CST IV under low supply. CST classifications are shown as a function of productivity due to shared (x-axis; $\overline{\tau}(1 - \chi)$) and private resources (y-axis; $\overline{\tau}\chi$), where $\overline{\tau}$ denotes the overall productivity. The grey diagonal line indicates an equal supply of the two resource types. The derivations of $\chi$ and $\overline{\tau}$ are found in the Extended Methods in Sect 1 in S1 Text. Sensitivity analyses of these results to different model assumptions are provided in Sect 4 in S1 Text. All the data and scripts used to generate the figure can be accessed at https://doi.org/10.57745/3GJF2Z.

Visualising the CSTs over the productivity owing to the distinct resource types corroborates that the relative supply of shared versus private resources has strong explanatory power for a large window of the resource landscape—i.e., at intermediate levels of sialic acid supply —($10^{-3} \lesssim \bar{\tau}\chi \lesssim 0.1$; Fig 3b). In this regime, either increasing the supply of shared resources or decreasing the supply of private resources is associated with a reduction in CST IV, most notably in favour of *L. iners*-dominated CST III, which has been hypothesised to act as a functional gateway between CST I and CST IV [17]. While resource-ratio theory—centred on relative resource supply—captures much of the observed variation in CST composition, deviations emerge at the extremes of the resource landscape. In particular, CST I and its functional analogues II and V are rarely found at high sialic acid supply (beyond $\bar{\tau}\chi = 10^{-2}$) while CST IV is rarely found at low sialic acid supply (below $\bar{\tau}\chi = 10^{-4}$) (Fig 3b). These edge cases suggest that, at the extremes of sialic acid supply, community composition may be driven primarily by sialic acid availability alone.

We also find that transitions between vaginal microbiota CSTs within individual women closely track changes in the proportion of resource supply private to CST IV, $\chi$ (Fig 4). A linear mixed-effects model with patient identity as a random effect shows that when two samples from the same woman were collected within seven days and belonged to the same CST (i.e., community persistence), the difference in $\chi$ was minimal (average $\Delta\ln(\chi)$ = 0.02, p-value = 0.84). In contrast, transitions from CST III to CST IV were associated with a significant increase in $\chi$ (average $\Delta\ln(\chi)$ = 1.36, p-value < 0.001) (Fig 4), supporting the idea that private resources such as sialic acid promote CST IV communities. Conversely, transitions away from CST IV (to CST I or CST III) tended to coincide with decreases in $\chi$ (Fig 4), although this trend was not statistically significant (average $\Delta\ln(\chi)$ = -0.35, p-value = 0.20).

## Discussion

Community ecology has long been proposed as a useful framework for understanding human microbiota dynamics. Our theoretical and empirical findings provide compelling evidence that resource ecology is a key factor shaping the composition and structure of the vaginal microbiota. While previous studies have demonstrated that elevated glycogen levels favour lactobacilli [18] and that sialidase activity affords *G. vaginalis* access to resources not accessible to lactobacilli [21], our findings bring into focus the ecological mechanisms linking the resource landscape to the microbiota composition. Specifically, we demonstrate that CST IV communities depend on relatively high levels of private resources to gain a growth advantage over CST I, which otherwise creates a hostile environment for CST IV—for example, through the production of lactic acid. As a consequence, across much of the resource landscape, either increasing the supply of private resources or decreasing the shared resources would shift the balance in favour of CST IV.

Our findings highlight the potential role of the vaginal microbiota's resource environment in informing both preventative and therapeutic strategies. Specifically, they support the idea that interventions that increase glycogen availability in the vagina may promote the growth of a *Lactobacillus*-dominant microbiome. Indeed, some BV treatment and prevention products are already marketed to leverage this dual mechanism: lactic acid helps restore vaginal pH, while glycogen serves as a nutrient source for *Lactobacillus* species. Conversely, strategies that limit access to private resources such as sialic acid may also help maintain or re-establish a *Lactobacillus*-dominated community. To reduce the bioavailability of private resources, several potential approaches could be considered, though their safety and feasibility remain unclear. These include breaking down sialic acids through lyase activity or lowering their effective concentration by polymerising them (i.e., into polysialic acid) or incorporating them into other structures, such as protein-glycans or lipid-glycans on human or bacterial cell surfaces. Furthermore, vaginal carrageenan treatment, which may have a preventive effect against HPV acquisition [36], could promote flushing out metabolites, potentially limiting sialic acid availability. However, this approach would need to affect sialic acids disproportionately, since overall resource production (our parameter $\tau$) has little effect on which bacteria dominate. Finally, there may be heterogeneity in microbiota resource ecology at the strain

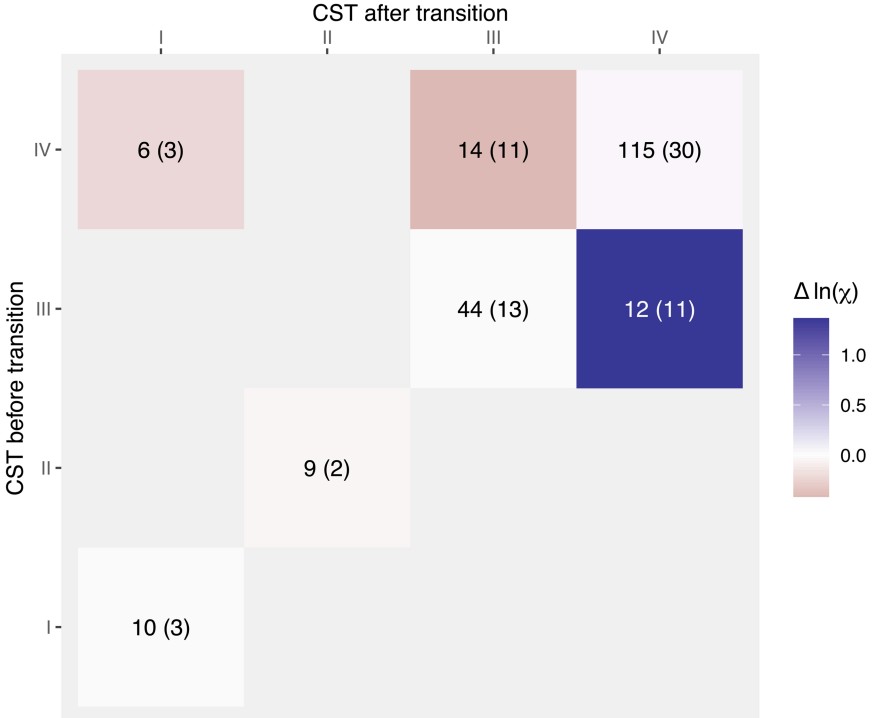

**Fig 4**. **Transitions between vaginal microbiota community state types (CST) are associated with changes in the proportion of supplied resource private to CST IV, $\chi$.** The estimate of private resource input, $\chi$, was informed by metabolomics and microbiological data. The difference ($\Delta$) in log-transformed $\chi$ was calculated between the next and focal observation. The number of observations (and patients) corresponding to each CST combination is shown in each tile. Successive observations separated by less than 7 days were analysed, as those with longer intervals were considered non-informative for CST transitions, which can take place over short periods. There were no transitions originating from CST I in this dataset, as the original cohort primarily enrolled women with symptomatic BV [32]. All the data and scripts used to generate the figure can be accessed at https://doi.org/10.57745/3GJF2Z.

level, with some *L. crispatus* strains being able to utilise sialic acid (albeit with low efficiency), while some (rare) *G. vaginalis* lack this ability. Therefore, optimal resource manipulation may require personalisation to the community composition at the strain level.

Supplementing private resources to help beneficial microbiota has been proposed in the context of improving gut microbiota health [37]. To date, it remains an open question if there are private resources for lactobacilli in the vaginal milieu. A recent study shows that these bacteria can hydrolyse innate immune peptides into free amino acids and metabolise them for their growth [38]. However, the relative importance of this additional source and the extent to which CST IV-associated bacteria engage in similar metabolic feats remain an open question. A search for lactobacilli-private resources warrants further inquiry, as our model demonstrates that private resource availability is a major force shaping the vaginal microbiota.

Mechanistic ecological modelling of vaginal microbiota is in its infancy [25,39]. Thus, there are ample opportunities for the future development of system-specific details. For example, one could explicitly model *L. iners* (CST III), whose unique features (i.e., it has a small genome size and carries a cytotoxin, inerolysin) and role as a transitional, or gateway taxon between CST I and IV has been increasingly recognised in recent years [11]. Even within CST I and IV, there are known differences between lineages. For example, some *L. crispatus* strains can metabolise sialic acid, when most cannot. Therefore, incorporating within-CST diversity into our model could yield equally interesting results as increasing the number of CSTs.

Another notable simplifying assumption in the present work is the omission of the immune system, despite its increasingly recognised role in shaping the vaginal microbiota [40]. Recent studies have shed light on the interactions between immune responses and microbial composition in the vaginal environment [41–43]. The direction of causality, however, remains unclear—does the microbiota shape the immune profile, or does immune signalling drive microbial changes? In all likelihood, both processes occur, but the relative magnitude and context-dependence of their effects remain insufficiently understood [17,44]. Characterising these signalling pathways and summarising the net immune response to various constituents of the vaginal milieu will be a key challenge in the future.

Ecological details of the vaginal milieu are often tentatively known, and variability among women and vaginal microbiota is rarely characterised quantitatively. Yet, our local sensitivity analyses (Sect 2 in S1 Text) demonstrate that details of within-host ecology—including resource supply and acquisition, relative growth rates, and competition as well as external microbial input—play a key role in shaping the structure of the human vaginal microbiota. Specifically, the factors identified to promote CST IV communities included: low productivity of the vaginal microbiota, sialidase activity efficiency, high CST IV-associated bacteria growth rate, low lactic acid production (slow acidification affords increased competition for resources) and low antagonism from CST I (Sect 2 in S1 Text). Fully parameterised with relevant data, mechanistic models would be able to generate quantitative predictions of the vaginal microbiota dynamics and compositions. Additionally, these models could facilitate *in silico* experiments that explore the consequences of interventions and counterfactual scenarios. To gain a systematic understanding of the ecological diversity of the vaginal microenvironment, the recent development of biomimetic *in vitro* models of vaginal epithelium offers a promising avenue [45]. Biomimetic *in vitro* models enable the cultivation and manipulation of reconstructed vaginal microbiota sourced from biological samples, providing a high-throughput approach for cataloguing the diversity of vaginal microbiota across different contexts.

While our work emphasises the pivotal role of the resource landscape in shaping the vaginal microbiota communities, it is worth noting other potential drivers. Factors such as the menstrual cycle, pregnancy, obesity, alcohol and tobacco use, antibiotics, ethnicity, stress, and the use of certain vaginal hygiene products have been implicated [17,46–50]. These factors may alter the physico-chemical conditions of the vaginal milieu (e.g., the formation of the vaginal mucus plug during pregnancy), while others (e.g., drug use) may also influence sexual behaviour, which in turn affects the exposure to BV-associated bacteria and sexually transmitted infections. To advance our mechanistic understanding of community transitions in real-world settings, mathematical models can be leveraged to interrogate how the influence of certain factors—for example, antibiotics that target certain bacteria—translates to the dynamics of vaginal microbiota communities.

A key advantage of the development of a mathematical model is to shed light on knowledge gaps in the literature, which future empirical studies may address. For example, the two resource types in our model are assumed to be perfectly substitutable, i.e., offer equal value to growth. Given that glycogen—a shared universal resource—is freely available in the extracellular milieu while sialic acid molecules—private resource only accessible to CST IV-associated bacteria—require activity by sialidases that cleave sialic acid from sialoglycans associated with mucins with potential handling time by the bacteria, it is conceivable that glycogen-fueled growth may be more energy efficient for CST IV-associated bacteria. Here again, *in vitro* experiments could study the growth rate of CST IV-associated bacteria under varied nutritional combinations and identify optimal nutritional requirements.

In addition, our mechanistic model integrates both resource production and consumption, enabling the role of the private resource to be inferred indirectly through modelling. Future studies employing direct sialidase activity assays, short-interval longitudinal sampling, or *in vitro* perturbations would enable a more direct empirical assessment of the causal mechanisms governing sialic acid dynamics. Such positive feedback between mathematical model development and targeted experiments will facilitate the development of model-assisted prognostic tools and evidence-based treatment options.

## Materials and methods

### Mechanistic ecological model of vaginal microbiota

At the fundamental level, our model can be interpreted as a consumer-resource model [51] in which $n$ communities of consumers ($C_i$, here bacteria) and their resource ($R$) are described by a set of ordinary differential equations:

$$\frac{dC_i}{dt} = -\gamma_C\, C_i + \lambda_i \frac{R}{\kappa + R} C_i - \alpha_{j,i}\, C_j\, C_i \tag{1a}$$

$$\frac{dR}{dt} = \eta_R - \gamma_R\, R - \sum_{i=1}^{n} \lambda_i \frac{R}{\epsilon(\kappa + R)}\, C_i \tag{1b}$$

where the growth of consumer type $i$ is determined by the maximum intrinsic growth rate ($\lambda_i$), conditioned upon the availability of the resource ($R$) following a Monod (or Holling type II) function ($R \mapsto {}^{R}\!/\!_{\kappa+R}$ where $\kappa$ represents the half-saturation constant, i.e., resource needed to achieve half the maximum growth). The resource is supplied to the system at a constant rate ($\eta_R$) and the resource and consumers exit the system at a background rate of $\gamma_R$ and $\gamma_C$, respectively. The resource is consumed by $C_i$ at a rate $\lambda_i {}^{R}\!/\!_{\epsilon(\kappa+R)}$ where $\epsilon$ corresponds to the yield (i.e., units of resource consumed per bacterial growth). Unlike classic consumer-resource models (e.g., [27]), in which the dynamics of a species only depend on its own density and that of the resources, we consider inhibitory interactions between two species through an $\alpha_{j,i}$ term capturing the impact of type $j$ on $i$. Note that more generic versions of similar formulation, but without Holling type II functions, have been developed to study microbial interactions [52], in which the resources correspond to interaction molecules such as bacteriocins or siderophores. Our work develops this generic model (Eq 1) focusing on two aspects of vaginal microbiota that functionally distinguish CST I from CST IV, namely differential resource use and lactic acid production with its inhibitory effect on bacterial populations.

a) *Differential resource use.* We model the availability of unbound sialic acid (i.e., private resource to CST IV) as a function of bacteria-independent resource supply $\eta_{Rp}$ and the density of CST IV bacteria that cleave off sialic acid from mucus sialoglycan. Thus, we express the realised input of CST IV private resource as $\eta_{Rp} {}^{C_4}\!/\!_{\theta + C_4}$ where $\theta$ represents the half-saturation constant of sialic acid cleaving. There is considerable variability in CST IV bacteria to cleave off and utilise sialic acid from bound sources [21]. In our model, small $\theta$ represents strong sialidase activity, where the majority of bound sialic acid can be made available by a few CST IV bacteria. We carry out sensitivity analyses to explore the impact of $\theta$ (Sect 4 in S1 Text).

b) *Inhibitory interactions mediated by lactic acid and other molecules.* Acidity is a key abiotic factor regulating microbial ecology in the vaginal environment [53]. The acidity is primarily maintained by lactobacilli through the production of lactic acid during the fermentation of glycogen and its derivatives that fuel the majority of their growth [54]. When dominated by lactobacilli, the vaginal milieu remains acidic with the pH ranging between 3.5 and 4.5. Acidity differentially affects bacterial communities. CST IV bacteria are commonly associated with vaginal pH >4.5 as their growth is inhibited at a lower pH [8,29]. In contrast, lactobacilli have a higher tolerance to acidic conditions (with an inhibition pH below 4.0 [29,30]). In our model, the lactic acid-mediated inhibitory effect on CST I and CST IV are denoted by $\alpha_{1,1}$ and $\alpha_{1,4}$, respectively, where $\alpha_{1,4} > \alpha_{1,1}$ because lactobacilli are more tolerant to acidic condition [17]. We note that the inhibition rate $\alpha_{1,4}$ also encompasses other mechanisms through which CST I inhibits CST IV bacteria including hydrogen peroxide ($H_2O_2$) and bacteriocins [17]. We ignore the reciprocal impact of CST IV to CST I ($\alpha_{4,1}$) as co-culture experiments have shown that *G. vaginalis* (CST IV) has negligible effects on *L. jensenii* (CST V, which is functionally akin to CST I) [55]. Several CST IV bacteria have evolved defence mechanisms including biogenic amines synthesis, which elevates the extracellular pH and alleviates the acidic stress for themselves as well as other bacteria [31,56]. Thus, $\alpha_{1,4}$ includes the net effect of inhibition by CST I and counteracting defence mechanisms of CST IV.

Our vaginal microbiota model (Fig 1) is described by the following set of equations:

$$\frac{dC_1}{dt} = -\gamma_C \, C_1 + \frac{\lambda_1 \, R_s \, C_1}{\kappa + R_s} - \alpha_{1,1} \, C_1^2 \tag{2a}$$

$$\frac{dC_4}{dt} = -\gamma_C \, C_4 + \frac{\lambda_4 \, (R_p + R_s) \, C_4}{\kappa + R_p + R_s} - \alpha_{1,4} \, C_1 \, C_4 \tag{2b}$$

$$\frac{dR_p}{dt} = \eta_{R_p} \frac{C_4}{\theta + C_4} - \gamma_R \, R_p - \frac{\lambda_4 \, R_p \, C_4}{\epsilon(\kappa + R_p + R_s)} \tag{2c}$$

$$\frac{dR_s}{dt} = \eta_{R_s} - \gamma_R \, R_s - \frac{\lambda_1 \, R_s \, C_1}{\epsilon(\kappa + R_s)} - \frac{\lambda_4 \, R_s \, C_4}{\epsilon(\kappa + R_p + R_s)} \tag{2d}$$

The subscripts 1 and 4 refer to CST I and CST IV bacteria and the subscripts $p$ and $s$ refer to private and shared resources, respectively.

## Model restructuring

To facilitate model exploration, we first restructure the above model (Eq 2) into equations tracking the dynamics of the total bacterial ($C = C_1 + C_4$) and resource ($R = R_p + R_s$) densities and proportions that make up the CST IV bacteria ($q = C_4/C$) and their private resource ($p = R_p/R$). We focus our analysis on the relative quantities, $p$ and $q$, in order to reduce the dimensionality of the original system—which involves two consumers and two resources—to a more tractable, one-dimensional framework. By working with relative rather than absolute quantities, we simplify the analysis without losing key ecological insights, allowing for a clearer and more tractable exploration than would be possible with the full two-by-two system. We then introduce $\chi$, the proportion of newly supplied resource that is private to $C_4$ (such that $\eta_{R_p} = \chi \, \eta_R$ and $\eta_{R_s} = (1 - \chi) \, \eta_R$ where $\eta_R$ is the overall supply). We also define $\psi$ as the ratio of CST IV inhibition and self-inhibition (i.e., $\alpha_{1,4}/\alpha_{1,1}$).

We then set the time scale of the system relative to the CST I growth rate ($\lambda_1$) such that selected parameters become relative to it and variables are dimensionless (i.e., non-dimensionalisation). Next, we assume a separation of timescales between consumers and resources and solve (owing to the Tikhonov-Fenichel theorem) for quasi-equilibrium conditions for resources such that the resource availability is held constant for a given density of consumers. In doing so, we now only explicitly track the consumer dynamics. In resource–consumer models, separating timescales is biologically justified when the dynamics of one variable—such as resource availability—occur on a much faster timescale than others, like consumer population changes. This reflects real biological systems, where resource concentrations can fluctuate rapidly due to environmental inputs or microbial uptake, while their assimilation into biomass (i.e., bacterial growth) typically proceeds more slowly, constrained by intracellular metabolic processing and regulation. By leveraging this disparity, time-scale separation simplifies model analysis and captures essential system behaviour without sacrificing biological realism.

Details about non-dimensionalisation can be found in the Extended Methods.

## Clinical data

We took advantage of the previously published human vaginal microbiota data from a longitudinal clinical study [32] and previously unpublished metabolomics data from the same cohort to parameterise the mechanistic ecological model and evaluate model predictions.

In particular, we could measure proxies for four of our variables. For the total amount of bacteria ($C$), the proxy was the ratio between the total bacterial DNA quantity in a sample and the median total bacterial DNA quantity in each of the samples using the qPCR data. For the proportion of CST IV bacteria ($q$), we used the proportion of CST IV-associated bacteria (here, the ones listed by the CST classification algorithm, VALENCIA [9]) in the 16S metabarcoding data. For the total

amount of resources ($R$), we used the sum of the abundance of four nutrients (glucose, maltose, maltotriose, and sialic acid) in a sample normalised by the median of these sums in all the samples using the metabolomics data. Finally, for the proportion of sialic acid among the resources ($p$), we used the ratio between the abundance of sialic acid and that of the four nutrients in the metabolomics data.

Details about the interfacing between the analytical expressions of the model and this data to parameterise the model can be found in the Extended Methods. The clinical study protocol was approved by the Institutional Review Board of the Johns Hopkins University School of Medicine and the University of Maryland School of Medicine. Written informed consent was appropriately obtained from all participants, which included permission to use the samples obtained in future studies.

**Participants and sample collection.** Briefly, the study—originally published in Ref [32]— focused on investigating bacterial vaginosis and included 135 women from Birmingham, Alabama, USA. At the beginning of the study, participants underwent an assessment that included an extensive questionnaire on their medical, dental, obstetric, hygiene, sexual, and behavioural histories. They were also asked to self-collect three mid-vaginal swabs daily for ten weeks for use in genomic sequencing and metabolomics analysis. In addition, participants recorded their daily hygienic practices and sexual activities in a diary using a standardised form. Pelvic examinations were conducted at the time of enrolment and at weeks 5 and 10 or at interim times if the participant reported vaginal symptoms. Any symptomatic BV was diagnosed by the clinician and treated using a standard of BV care practices [57]. Integration with our mechanistic model requires a sample to be processed for both genomic sequencing (16S metabarcoding and qPCR DNA quantity) and resource-related metabolomics profiling. In total, 540 samples from 82 women met our criteria.

**Microbiota data:** DNA was extracted from vaginal swabs and then the V3-V4 region of the bacterial 16S rRNA gene was amplified and sequenced on an Illumina HiSeq 2500 as described previously [58]. The paired end sequences were then processed using DADA2 [59] generating amplicon sequence variants (ASVs). The ASVs were then assigned taxonomy using a combination of the RDP Naive Bayesian Classifier [60] trained on the SILVA 16S rRNA gene database [61] and speciateIT software [62]. Finally, community state types (CSTs) were assigned using VALENCIA software [9]. The total bacterial DNA was quantified using qPCR. We normalised the bacterial DNA quantity with the median total qPCR quantity of all samples such that the default value ($C = 1$) corresponds to the study median as explained above. These data have been published in Ref. [32].

**Metabolomics data:** As indicated above, we also provide original data on metabolites from the same vaginal swabs. These were processed by Metabolon for global liquid chromatography-mass spectrometry (LC-MS) metabolomic profiling, which provides information roughly proportional to the true concentration of the metabolites in a sample [63]. Metabolomics readings were batch-standardised by Metabolon. An important limitation of global metabolomic profiling is that it provides an imperfect reflection of the metabolite quantity in a sample. Specifically, the Pearson's correlation coefficient between the raw global metabolomics output and concentrations measured by the targeted metabolomics approach (which offers the most accurate quantitative information) is generally around 0.7 [64]. As metabolites are small molecules less than 1.5 kDa in size [65], whole glycogen molecules are not detected by metabolomics profiling. Thus, we focused on three glycogen breakdown products (namely, glucose, maltose, and maltotriose) as a proxy for the shared resource. We refer to the sum of these three metabolites as glycogen derivatives. As a proxy for the private resource used by CST IV, we used N-acetylneuraminate, the anionic form of the acetylneuraminic acid (Neu5Ac), which is the form of sialic acid found in the vaginal environment [21]. Fucose, another private resource for CST IV, was not found in the metabolomics panel. With the aforementioned caveat that global metabolomic profiling provides an imperfect reflection of metabolite quantity, we use resource-related metabolomics data in two ways: total resource-related metabolite quantity (i.e., the sum of glucose, maltose, maltotriose and sialic acid) and proportion of CST IV private resource (i.e., sialic acid relative to the total resource quantity). We normalised raw resource metabolite values by the total resource-related metabolites across all samples such that the default total resource value ($R = 1$) corresponds to the study median.

## Evaluating model predictions in light of clinical data

To corroborate the role of resources in shaping real vaginal microbiota communities, we graphically explored the relationship between the empirically informed proportion of CST IV private resources ($\chi$; see eq. 8b in S1 Text) and CST classification. We also visualised CST classification as a function of shared productivity (i.e., yield-adjusted glycogen derivative supply, i.e., $\overline{\tau}(1-\chi)$) and private productivity (i.e., yield-adjusted sialic acid supply, $\overline{\tau}\,\chi$) to explore the influence of the absolute resource supply in shaping bacterial community composition. As the half-saturation constants (i.e., $\kappa$ and $\theta$) are not precisely informed by data, we do not rely on the absolute value of $\overline{\tau}$ and $\chi$. Nonetheless, our sensitivity analyses demonstrate that the rank order of $\overline{\tau}$ and $\chi$ estimates are largely consistent across a range of $\theta$ and $\kappa$ such that our findings are qualitatively unaffected except in the case of very large $\theta$ such that private resources are virtually inaccessible (Sect 4 in S1 Text).

The association between changes in the supply of private resources and community transitions was explored visually, and using a linear mixed-effects model with patient identity as a random effect. The difference ($\Delta$) in log-transformed $\chi$ was computed between each focal observation and the subsequent observation. Only successive observations separated by fewer than 7 days were included, as longer intervals were considered non-informative for CST transitions, which can take place over short time spans.

## Acknowledgments

Version 4 of this preprint has been peer-reviewed and recommended by Peer Community In Ecology (https://doi.org/10.24072/pci.ecology.100777).

## Supporting information

**S1 Text. The supporting information S1 Text contains i) Extended Methods, ii) Local sensitivity of qualitative outcomes to model parameters, iii) Frequency of vaginal bacteria species as a function of private resource input, and iv) Sensitivity to half-saturation constants.** The works cited exclusively in the Supporting information are as follows:[66–73]. All the data and scripts used to generate the figures can be accessed at https://doi.org/10.57745/3GJF2Z. (PDF)

## Author contributions

**Conceptualization:** Tsukushi Kamiya, Carmen Lia Murall, Jacques Ravel, Samuel Alizon.

**Data curation:** Tsukushi Kamiya, Michael France.

**Formal analysis:** Tsukushi Kamiya, Michael France.

**Funding acquisition:** Samuel Alizon.

**Investigation:** Tsukushi Kamiya, Nicolas Tessandier, Ignacio G Bravo, Samuel Alizon.

**Methodology:** Tsukushi Kamiya, Mircea T Sofonea, Samuel Alizon.

**Supervision:** Samuel Alizon.

**Validation:** Mircea T Sofonea, Nicolas Tessandier, Ignacio G Bravo, Carmen Lia Murall, Jacques Ravel, Samuel Alizon.

**Visualization:** Tsukushi Kamiya.

**Writing – original draft:** Tsukushi Kamiya, Samuel Alizon.

**Writing – review & editing:** Tsukushi Kamiya, Mircea T Sofonea, Michael France, Nicolas Tessandier, Ignacio G Bravo, Carmen Lia Murall, Jacques Ravel, Samuel Alizon.

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
