## [Editor Report · Decision Letter 0]

10 Oct 2025

Dear Dr Alizon,

Thank you for submitting your reviewed and revised PCI-Ecology manuscript entitled "Resource landscape shapes the composition and stability of the human vaginal microbiota" for consideration as a Research Article by PLOS Biology.

Your manuscript has now been evaluated by the PLOS Biology editorial staff, as well as by an academic editor with relevant expertise, and I'm writing to let you know that we would like to consider your submission further.

However, before we can do so, we need you to complete your submission by providing the metadata that is required for full assessment. To this end, please login to Editorial Manager where you will find the paper in the 'Submissions Needing Revisions' folder on your homepage. Please click 'Revise Submission' from the Action Links and complete all additional questions in the submission questionnaire.

Once your full submission is complete, your paper will undergo a series of checks in preparation for further consideration. To provide the metadata for your submission, please Login to Editorial Manager (https://www.editorialmanager.com/pbiology) within two working days, i.e. by Oct 14 2025 11:59PM.

Kind regards,

Roli Roberts

Roland Roberts, PhD

Senior Editor

PLOS Biology

rroberts@plos.org

---

## [Editor Report · Decision Letter 1]

13 Oct 2025

Dear Dr Alizon,

Thank you for your patience while we considered your revised PCI Ecology manuscript "Resource landscape shapes the composition and stability of the human vaginal microbiota" for consideration as a Research Article at PLOS Biology. Your revised study has now been evaluated by the PLOS Biology editors and the Academic Editor.

In light of the Academic Editor's comments, which you will find at the end of this email, we are pleased to offer you the opportunity to address these remaining points in a revision that we anticipate should not take you very long. We will then assess your revised manuscript and your response with our Academic Editor.

IMPORTANT - please attend to the following:

a) Please address the comments from the Academic Editor (see the foot of this email).

b) We thought that your paper would be better considered as a Short Report. I’ve changed the article type accordingly, but there is no need for further re-formatting.

c) Many thanks for providing the necessary data and code. Please cite its location clearly in all relevant main and supplementary Figure legends (Figs 2ABCD, 3AB, S1, S2, S3, S4, S5), e.g. “The data and code needed to generate this Figure can be found at https://doi.org/10.57745/3GJF2Z”

d) Your Financial Disclosure statement says “The author(s) received no specific funding for this work.” However, in your Acknowledgements, you say "The authors acknowledge support from the Fondation pour la Recherche Medicale (SPF202005011951to TK), the Centre National de la Recherche Scientifique, the Agence Nationale de la Recherche contre le Sida (ANRS—MIE, to NT), the US National Institute of Health (NR015495 to JR)." Please transfer this information into the metadata on the submission form.

e) Please include the following wording in your Acknowledgements, in order to recognise the contribution of PCI: “Version 4 of this preprint has been peer-reviewed and recommended by Peer Community In Ecology (https://doi.org/10.24072/pci.ecology.100777).”

f) Please ensure that the references in your supplement are also present in your main references, to ensure that due credit is received.

**IMPORTANT - SUBMITTING YOUR REVISION**

*Resubmission Checklist*

*Published Peer Review*

*PLOS Data Policy*

Sincerely,

Roli Roberts

Roland Roberts, PhD

Senior Editor

PLOS Biology

rroberts@plos.org

COMMENTS FROM THE ACADEMIC EDITOR:

I would like to ask the authors the following:

- As a non-expert, I was wondering if there would also be a possibility to reduce the private resource in real interventions rather than providing more of the shared resource(s). I may have missed it, but this possibility does not seem to be discussed.

- I found the paragraph starting from l. 176 a bit awkwardly phrased. From a mathematical standpoint, it is obvious that once one consumer is lost, it cannot reappear except via "seeding". I think that the authors want to highlight that the bistability leads to different amounts of seeding that are necessary to bring the system into the basin of attraction of the other stable state, but the phrasing made me think of a trivial statement at first.

- A major appeal of this paper is that the authors used data to evaluate their predictions by developing ways to break down the real-life systems into simple parameters. However, I was wondering why they decided to use the dominant group as a metric in figure 3, rather than focusing on the proportions of CSTIV like in the model predictions shown in figure 2. Moreover, I would be very curious to hear whether the authors found signals of their predictions in individual time series when the community shifted in the same woman over time.

---

## [Editor Report · Decision Letter 2]

8 Dec 2025

Dear Samuel,

Thank you for the submission of your revised Short Reports "Resource landscape shapes the composition and stability of the human vaginal microbiota" for publication in PLOS Biology. On behalf of my colleagues and the Academic Editor, Claudia Bank, I'm pleased to say that we can in principle accept your manuscript for publication, provided you address any remaining formatting and reporting issues. These will be detailed in an email you should receive within 2-3 business days from our colleagues in the journal operations team; no action is required from you until then. Please note that we will not be able to formally accept your manuscript and schedule it for publication until you have completed any requested changes.

IMPORTANT: I've asked my colleagues to include the following two requests among their own: 1. Please include the URLs of your funders in the Financial Disclosure statement. 2. We noticed a typo in line 281 of your manuscript: "while some (rare) G. vaginalis lacking this ability" should be either "while some (rare) G. vaginalis lack this ability" or "and some (rare) G. vaginalis lacking this ability."

PRESS : We frequently collaborate with press offices. If your institution or institutions have a press office, please notify them about your upcoming paper at this point, to enable them to help maximise its impact. If the press office is planning to promote your findings, we would be grateful if they could coordinate with biologypress@plos.org. If you have previously opted in to the early version process, we ask that you notify us immediately of any press plans so that we may opt out on your behalf.

Sincerely,

Roli

Senior Editor

PLOS Biology

rroberts@plos.org